



**Uncertainty in counting ice nucleating particles with continuous**
**diffusion flow chambers**
**Sarvesh Garimella[1], Daniel A. Rothenberg[1], Martin J. Wolf[1], Robert O. David[2], Zamin A.**
**Kanji[2], Chien Wang[1], Michael Rösch[1], and Daniel J. Cziczo[1,3]**
[1] {Department of Earth, Atmospheric and Planetary Sciences, Massachusetts Institute of
Technology, Cambridge, MA, United States}
[2] {Institute for Atmospheric and Climate Science, ETH Zurich, Zurich, Switzerland}
[3] {Department of Civil and Environmental Engineering, Massachusetts Institute of
Technology, Cambridge, MA, United States}
Correspondence to: D. J. Cziczo (djcziczo@mit.edu)



**Abstract**
This study investigates the measurement of ice nucleating particle (INP) concentration and sizing
of crystals using continuous flow diffusion chambers (CFDCs). CFDCs have been deployed for
decades to measure the formation of INPs under controlled humidity and temperature conditions
in laboratory studies and by ambient aerosol populations. These measurements have, in turn,
been used to construct parameterizations for use in models by relating the formation of ice
crystals to state variables such as temperature, humidity, and aerosol particle properties such as
composition and number. We show here that assumptions of ideal instrument behavior are not
supported by measurements made with a commercially available CFDC, the SPectrometer for
Ice Nucleation (SPIN), and the instrument on which it is based, the Zurich Ice Nucleation
Chamber (ZINC). Non-ideal instrument behavior, which is likely inherent to varying degrees in
all CFDCs, is caused by exposure of particles to different humidities and/or temperatures than
predicated from theory. This can result in a systematic, and variable, underestimation of reported
INP concentrations. We use a machine learning approach to show that non-ideality is most likely
due to small scale flow features where the aerosols are combined with sheath flows and to
minimize the uncertainty associated with measured INP concentrations. We suggest that detailed
measurement, on an instrument-by-instrument basis, be performed to characterize this
uncertainty.



**1. Introduction**

2       Aerosol particles affect the climate system via their ability to interact with radiation and act

as the sites upon which water condenses to form liquid and ice clouds (Pruppacher and Klett,
1997; Seinfeld and Pandis, 2006; Boucher et al., 2013). Those that facilitate ice crystal formation
above the temperature or below the humidity of homogeneous freezing are called ice nucleating
particles (INPs) and these affect the formation and persistence of mixed-phase and cirrus clouds
(Forster et al., 2007). The interactions between INPs and cold clouds are a measurement
challenge because such clouds occur either high in the atmosphere or near the poles and are
difficult to access (Rossow and Schiffer, 1999). Continuous flow diffusion chambers (CFDCs)
have provided a means to understand ice cloud formation by measuring INP concentrations in
the field. By exposing an ambient aerosol population to controlled humidity and temperature
conditions, the ability of natural aerosols to activate as INPs can be quantified (DeMott et al.,
2003a; DeMott et al., 2003b; Chou et al., 2011; Boose et al., 2016).

14       Measurements of INP concentration using CFDCs have been used to construct model

parameterizations that relate the formation of ice crystals to temperature and aerosol particle
number and size (DeMott et al., 2010; Tobo et al, 2013; DeMott et al., 2015). Using such
parameterizations, global aerosol transport models attempt to link aerosol emissions to their
potential to impact ice cloud formation and thus climate (Vergara-Temprado et al., 2016). The
use of CFDC data for parameterization of ice formation in such models highlights the need for
accurate and unbiased measurements.

21       CFDC instruments are able to determine INP concentration by drawing in aerosol particles

and controlling the temperature and relative humidity to which they are exposed (Rogers, 1988;
Stetzer et al., 2008; Garimella et al., 2016). Although there are instrument-to-instrument



differences in geometry and flows, typically particles are drawn through an inlet and contained
between two sheath flows (Figure 1). These three flows pass between two ice-coated walls that
are held at different, sub-0º C, temperatures. Water vapor and heat diffuse from the warm wall to
the cold wall, such that approximately linear gradients of both quantities exist across the width of
the chamber. Because the saturation vapor pressure exhibits a nonlinear temperature dependence,
the air within the chamber is supersaturated with respect to ice. The variation in heat and vapor
diffusion results in a maximum in supersaturation near the center of the chamber (Rogers, 1988).
Particles constrained to a narrow central lamina by the sheath flows should, in theory, be
exposed to only the maximum saturation with a small uncertainty in temperature and humidity.
The fractional width of the lamina is typically taken to be the ratio of the incoming aerosol flow
rate to the total (sample + sheaths) flow rate through the chamber (Rogers, 1988).

12       A sufficiently large temperature gradient between the walls can cause lamina conditions to

not only exceed ice but also liquid water saturation (Rogers, 1988; Stetzer et al., 2008; Garimella
et al., 2016). Droplet formation is important since many CFDCs measure only the size of objects
exiting the chamber with an Optical Particle Sizer (OPS, Rogers, 1988; Demott et al., 2015). The
presence of droplets can therefore be misinterpreted as a higher abundance of ice crystals. The
impact of droplet formation is minimized by the utilization of an "evaporation region" in most
modern CFDCs. These regions are isothermal and ice-coated sections at the bottom of the
chamber where small droplets are evaporated by subsaturated conditions with respect to liquid
water. Nonetheless, a CFDC run at a sufficiently large temperature gradient between the walls
can create droplets large enough to survive evaporation sections. This sets a condition known as
"droplet breakthrough" that is specific to each CFDC's geometry and flow characteristics
(Rogers, 1988; Stetzer et al., 2008; Garimella et al., 2016).



Instruments rarely follow theoretical predictions. In the case of CFDCs, this is often due to
non-ideal flow conditions and deviations from isothermality. DeMott et al. (2015) discussed the
effect of aerosol "spreading" outside the lamina, and the resulting low bias in the number of INP
measured. Here, we extend the work of DeMott et al. (2015) with a quantitative analysis of the
source and effect of spreading and discussion of the impact on CFDC data. For this work we use
the Zurich Ice Nucleation Chamber (ZINC, Stetzer et al., 2008) and the commercial version, the
SPectrometer for Ice Nucleation (SPIN, Garimella et al., 2016). The automation of these
instruments, in particular the large amount of "housekeeping" data autonomously recorded to
characterize SPIN instrument behavior, makes these chambers suitable for exploring this effect.
We apply a machine learning algorithm for analysis in order to process the large amount of data
and generate statistical inferences to constrain the spreading effect. We suggest the spreading
effect can be best visualized as a deviation from laminar flow and non-isokinetic injection as the
particles are drawn into the chamber. We conclude that the non-ideal conditions are likely
universal but also dependent on the geometry and flow characteristics of each CFDC chamber.
**2. Methodology**
**2.1 Particle timing tests**

18       The ZINC and SPIN CFDCs have been described in detail previously (Stetzer et al., 2008;

Garimella et al., 2016). To measure the degree of particle spreading outside the lamina a precise
particle pulse was introduced into the chambers. In the case of SPIN this was a 1 second pulse
while for ZINC a 10 second pulse was used. In both cases a valve at the chamber inlet was used
to control the pulse. Under ideal conditions this should correspond to an equivalent particle pulse
at the chamber outlet. Non-idealities have been shown to lead to particle spreading across the





width of the chamber as they traverse its length by DeMott et al. (2015). For this work the
arrival of particles was measured at the chamber outlet with a Condensation Particle Counter
(Brechtel, Inc. CPC Model 1720 for SPIN, and TSI CPC 3772/3787 for ZINC). A wider particle
pulse (in time) measured at the outlet indicates more spreading of the particles across the width
of the chamber, since the fastest particles travel closer to the center of the chamber under a
laminar flow assumption. This is shown in Figure 2 for ZINC experiments at a total flow rate of
10 lpm and chamber conditions of -40º C and 102% relative humidity (RH, Panel A) and 110%
RH with respect to water (Panel B). 10 second pulses were produced with 200 nm ammonium
nitrate (Sigma Aldrich) particles which were wet-generated using an atomizer  and size selected
with a differential mobility analyzer (TSI DMA 3082). CPC measurement at the input (CPC$_{in}$)
verifies production of a 10-second pulse while the output particles (CPC$_{out}$) continue for 20 – 30
seconds.
The SPIN data exhibit the same behavior. SPIN particle distributions were measured for 30
1-second aerosol pulses at constant conditions of 20° C and ~10 lpm flow. 100 nm diameter
ammonium sulfate particles wet-generated and dried with a Brechtel Manufacturing, Inc. (BMI)
9203 Aerosol Generator and mobility diameter selected with a Brechtel, Inc. Differential
Mobility Analyzer Model 2100 were used. Combining the arrival pulse with the shape of the
velocity profile the corresponding distribution of particles across the width of the chamber can be
determined (Figure 3).
A further ~250 pulse measurements using ambient aerosol particles were conducted using the
SPIN setup at Storm Peak Laboratory (Steamboat Springs, Colorado, 3220 m M.S.L.; 40.455ºN,
-106.744ºW) to capture the spreading effect variability in an environment where INP field
measurement campaign occur (DeMott et al., 2003a). These tests were across a range of chamber



thermodynamic conditions (lamina humidities between ice and water saturation at temperatures -
15 to -40º C).

## 2.2 Machine learning prediction

A random forest regression (RFR) (Breiman, 2001) was used to predict the fraction of
particles that remained in the aerosol lamina (hereafter "$f_{lam}$"). In this application RFR is
similar to a multiple linear regression except that it grows a forest of bootstrap aggregated (or
"bagged") decision trees to fit the data instead of using a linear model. Bootstrap aggregation
avoids overfitting the data, provides uncertainty quantification for each prediction using the out-
of-bag (oob) prediction error, ranks the variables by their importance by comparing oob
prediction errors, and does not assume linear relationships between variables (Breiman, 2001).
First, the complete set of housekeeping variables recorded for SPIN is input to the RFR, for
which they are termed "features". This housekeeping data set is normally recorded to verify
instrument operation and no *a priori* assumptions are made as to which variables are the most
important predictor of $f_{lam}$; the RFR indicates the most import predictors by comparison to the
experimental pulse results. As an example, ambient temperature might not be expected to be an
important factor in the spreading effect but it was not removed from the data set; that decision
was left to the RFR. Feature importance was observed to fall exponentially and those within the
first two e-folding lengths of importance were maintained in a reduced RFR model. The reduced
RFR subset included 65 variables including wall temperature, flows, and thermodynamic
variables predominantly in the middle-top of the SPIN chamber (Garimella et al., 2016); this is
the region of the chamber where aerosol is encased within the sheath flows. The top ten most
important features are listed in Table 1.





**3. Results and Discussion**

3        Figure 4 shows the results from the 30 particle timing tests at 20° C and ~10 lpm flow

conditions. The fraction of particles that remained in the aerosol lamina varied despite constant
flow, aerosol properties, and temperature. Figure 5 shows the results from 267 ambient particle
pulse experiments in the aforementioned temperature and saturation range. In Figure 5, $f_{lam}$ is
plotted against the reported lamina temperature and ice saturation ratio ($S_{ice}$), the actively
controlled variables in CFDC chambers. Data are not highly correlated to either. The mean and
standard deviation of $f_{lam}$ are 0.25 ± 0.14 and, depending on the specific conditions, the
distribution exhibits values that vary between 0.03 and 0.73 (i.e., between 3 and 73% of particles
were within the lamina).

12       The reduced RFR described in Section 2.2 can be used to predict $f_{lam}$ (mean values and

standard deviations) based on the SPIN variables shown to be most important. Figure 6 shows
the performance of this approach, which has an oob mean squared prediction error of 0.008
whereas simply selecting the mean value for $f_{lam}$ from the distribution in Figure 5 results in a
mean squared error for predicting $f_{lam}$ of ~0.02; the RFR approach reduces the uncertainty by
~60%.

18       DeMott et al. (2015) noted the non-ideality of the Colorado State University CFDC chamber.

They expanded upon the work of Tobo et al. (2013) by proposing the addition of a "calibration
factor" ($cf$) = 3 by which the measured INP number could be multiplied to provide a corrected
value. By definition, Tobo et al. (2013) (and all previous studies) used $cf = 1$; this corresponds
to an assumption that all particles in a CFDC exist within the lamina, which DeMott et al. (2015)
showed to be incorrect. $cf = 3$ corresponds to a constant 33% of particles existing within the



lamina regardless of flow or thermodynamic state. The distribution found here corresponds to a
variable correction factor in the range of $cf = 2.6$ to 9.5, depending on the experiment, with a
mean of 4. We note the $cf = 3$ value reported by DeMott et al. (2015) is for a different CFDC
but falls within the range measured here. Our work does not, however, support a fixed $cf$ value,
at least for SPIN.

6        Further evidence to support the spreading effect is provided by the size of the ice crystals

measured at the output of a CFDC. Theoretically, a monodisperse population of an aerosol
composition that only nucleates ice homogeneously should exhibit freezing at the same time and
location within a CFDC chamber. This should translate to a monodisperse ice crystal size
distribution at the chamber output; the size of the crystals should be a function of the chamber
RH and temperature and equivalent to the amount of vapor-deposited water under these
conditions. The result of the extended time it takes for particles to exit the chamber due to
spreading would be (1) larger crystals due to extended time in a supersaturated region and (2) a
broadening of the ice crystals size distribution due to residence time and supersaturation
variability to which the particles are exposed. Experiments run with the ZINC chamber confirm a
non-monodisperse ice crystal size distribution (Figure 7). Aerosol particles were assumed to
nucleate ice immediately upon entering the chamber since the -40° C lamina temperature was
below that required for homogeneous ice nucleation and four cases are considered. The
combination of velocity profile and residence time from the pulse experiments (Figure 2) were
used to determine the location of the particles in the lamina and therefore the time they were
exposed to variable supersaturation and the subsequent size to which they would grow. The
baseline crystal size was when all particles remained within the predicated lamina (i.e., within
the dash lines in Figure 1) and is monodisperse at ~4 micrometers diameter (orange histogram).



When minor spreading in time with respect to that of that predicted when particles remained
within the lamina was allowed, at the level of +/- 1 second, the ice crystal distribution broadened
(3-4 micrometers diameter; magenta histogram). The extended time that ice crystals were
observed to remain in ZINC experiments (Figure 2) caused a further broadening in ice crystal
size distribution (3-7 micrometers diameter; blue histogram). The measured ice crystal size
distribution (yellow histogram) shows particles predominantly from 2-7 micrometers and is most
consistent with the calculations made for ice crystal growth that include the spreading effect.
Note that ice smaller than 3 micrometers diameter may be due to crystals that are undersized by
passing through the edge of the ZINC OPS (Stetzer et al., 2008).
The effect of particle spreading outside the lamina on CFDC reports of INP concentration
measurements can be visualized using the data collected here. Figure 8 shows idealized
activation curves (i.e., nucleation of ice or droplets) at various $f_{lam}$ values. Note that $f_{lam}$ and $cf$
can be thought of interchangeably where 33% and 3 are, respectively, equivalent. The aerosol
population is assumed to be "perfect" immersion mode INP that form ice crystals immediately
upon exposure at water saturation ($S_{liq}$=1); this could be viewed as a laboratory test of effective
immersion INPs. In the case where the CFDC is assumed to operate ideally, all particles are
constrained within the lamina ($f_{lam}$ = 100%) and all nucleation occurs at $S_{liq}$=1 (solid line).
The other three curves in Figure 8 correspond to increasingly less ideal behavior (i.e.,
increasingly fewer particles in the lamina), corresponding to $f_{lam}$ falling from 33 to 10%. The
deviation from the ideal case can be viewed as a higher than saturation condition at the centerline
required so the particle farthest outside the lamina experiences this value. These can also be
interpreted as cases where $cf$ is fixed but increases from 3, the value suggested by DeMott et al.
(2015), to 10, the worst case found in this work.



Figure 9 expands on Figure 8 by considering a case more applicable to measurement of an
ambient aerosol population. In this case only 10% of the particles are perfect immersion mode
INP, whereas the rest are cloud condensation nuclei (CCN) that activate at exactly $S_{liq} = 1$. The
evaporation section on the bottom of the heuristic chamber is equivalent to that of SPIN so that
droplets evaporate until breakthrough at $S_{liq} > 1.07$ (Garimella et al., 2016). In the ideal case all
particles are constrained within the lamina and 10% of particles nucleate ice at a CFDC
saturation of $S_{liq} = 1$ (panel a, solid black line). The remaining 90% of particles break through
as droplets at $S_{liq} > 1.07$ (panel a, solid blue line). The other three curves correspond to the
increasingly less ideal behavior presented in Figure 8, corresponding to $f_{lam}$ falling from 33 to
10%. In these cases an increasingly higher maximum saturation is required so that the particles
farthest from the centerline experience $S_{liq} = 1$ (ice nucleation) and 1.07 (droplet
breakthrough). The resulting activation curves if droplets and ice crystals are indistinguishable
(i.e., a composite of the black and blue traces in panel a), historically the case for CFDC
detectors (Rodgers, 1988), is shown in panel b. The shape of the idealized activation curve in
Figure 9b resembles that of experimental CFDC activation curves (DeMott et al., 2015) due to
the dependence of activated fraction on $S_{liq}$ because of the particle spreading effect.

## 18     5. Conclusions

The results presented here indicate that neither the reported thermodynamic conditions nor
results from a single timing test capture the full variability of $f_{lam}$ in the SPIN CFDC. Following
on the results of DeMott et al. (2015), the findings in this study indicate that $f_{lam}$ is not unity in
real CFDCs. We show that it is also variable in ZINC and SPIN. We believe this is likely
universal to all CFDC instruments although the degree of uncertainty and magnitude of the effect



are probably a function of instrument geometry, flow and thermodynamic conditions. The non-
uniform time particles spend in a CFDC has complex results on ice nucleation and crystal
growth, including larger and broader size distributions than predicted by theory.

4       A machine learning approach used housekeeping data to show that the most likely reason for

the lack of ideality is small scale flow features near the area of sheathing the aerosol sample
flow; the RFR deemed variables including wall temperature, flows, and thermodynamic
variables predominantly in the middle to top section of the SPIN chamber (i.e., at the injection
point) as most important. Moreover, the RFR approach was able to better predict $f_{lam}$, and
therefore the conditions experienced by the aerosols in the chamber than standard CFDC flow
theory with an overall reduction in uncertainty by $\sim$60%.

11       Finally, we show the particle spreading effect explains why CFDC chambers are often

operated at non-physical $S_{liq}$ values to measure immersion mode INP and why the reported
numbers are strongly dependent on $S_{liq}$. Theoretically, immersion mode nucleation should occur
at $S_{liq} = 1$, yet reports with CFDCs often show increased concentrations up to, and often well
beyond, 1.05. By contrast, CCN instruments routinely activate essentially all particles into
droplets at $1.01 - 1.02$.

17       We suggest laboratory work determining the extent of spreading variability be conducted for

all CFDC chambers to minimize this bias and its variability. We suggest this work would (1)
explore how experimental and chamber design influence the spreading effect, drawing
comparisons to computational fluid dynamics simulations to complement the RFR statistical
modeling and (2) which operational considerations (such as flow rates, inlet pressure drop, etc.)
maximize probability of isokinetic injection of particles into the chamber.





**Acknowledgements**
The authors gratefully acknowledge funding from NASA grant # NNX13AO15G, NSF grant #
AGS-1461347, NSF grant # AGS-1339264, and DOE grant # DE-SC0014487. Z. A. Kanji and
R. O. David would like to acknowledge funding from SNF grant # 200021_156581. We would
like to thank the Storm Peak Laboratory staff for the use of its facilities to perform the pressure-
dependent timing experiments. We thank Paul DeMott, Jesse Kroll and the Fifth Ice Nucleation
workshop participants for helpful discussions.





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





| Feature Rank | Feature Name |
|---|---|
| 1 | Lamina Saturation at TC3 |
| 2 | Average ΔT between Warm and Cold Wall |
| 3 | Lamina Saturation at TC4 |
| 4 | Warm Wall H0 On/Off Time |
| 5 | Lamina Saturation at TC1 |
| 6 | Lamina Saturation at TC11 |
| 7 | Average Warm Wall Temperature Spread |
| 8 | Lamina Saturation at TC5 |
| 9 | Average Warm Wall Temperature Difference from Set Point |
| 10 | Total Volume Flow at Mass Flow Controllers |

2    Table 1: List of the ten most important features from the RFR. Thermocouple (TC#) and heater

3    (H#) locations correspond to those described in Garimella et al. (2016). The features are

4    predominantly located in the top and middle sections of the chamber.





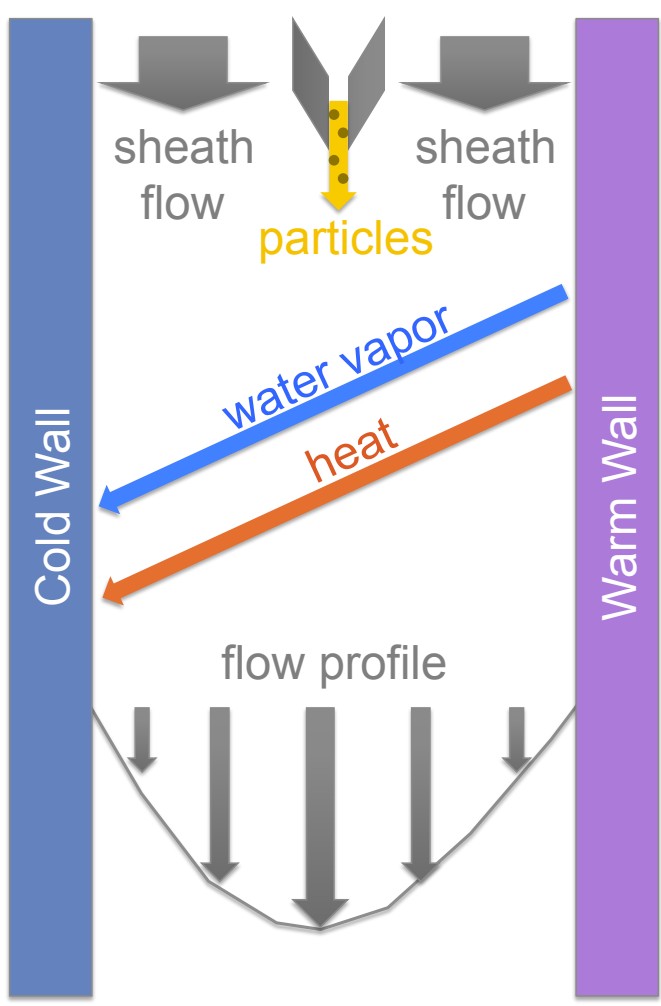

Figure 1: Schematic representation of an idealized CFDC. A particle-laden flow is passed
between two ice-coated walls that are held at different temperatures below 0ºC. This results in
water vapor and heat diffusing from the warm to the cold wall. Supersaturation, with a maximum
near the centerline, results from the non-linear relationship of water vapor saturation with respect
to temperature. Sheath flows along each wall are meant to isolate particles to a central lamina at
or near the supersaturation maximum, which also theoretically restricts the temperature and
superstation to which they are exposed.



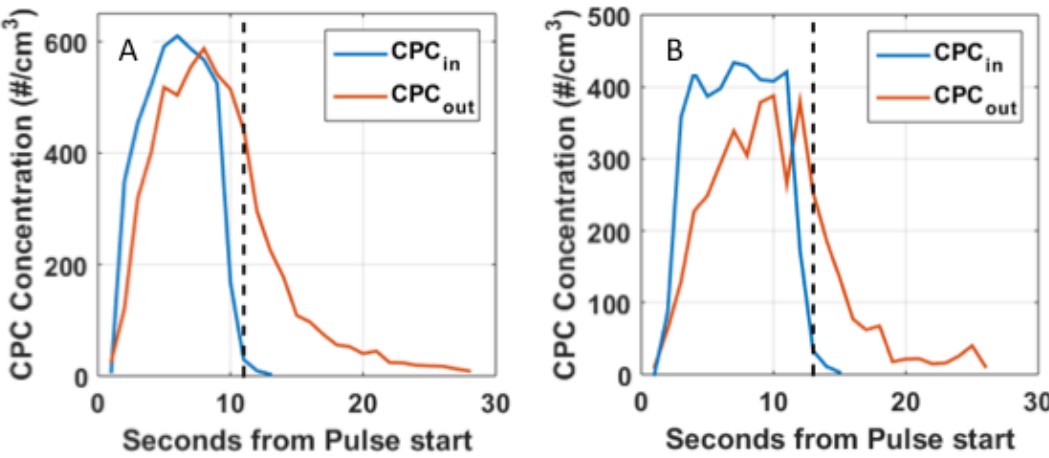

Figure 2. Particle concentration as a function of time for particle pulse experiments using ZINC.
The blue traces are the counts measured at the entrance of the chamber (CPC$_{in}$) while the red
traces are the concentrations at the output (CPC$_{out}$). Both experiments were conducted at -40° C
with Panel A at 102% RH and Panel B at 110% RH, both with respect to liquid water. Particles
in the red trace occurring after the vertical dashed line are outside the initial pulse duration and
are inferred to have moved out of the lamina. The ratio of particles within the pulse time at the
outlet versus the total particles were 77.7 and 76.2% for Panels A and B, respectively.





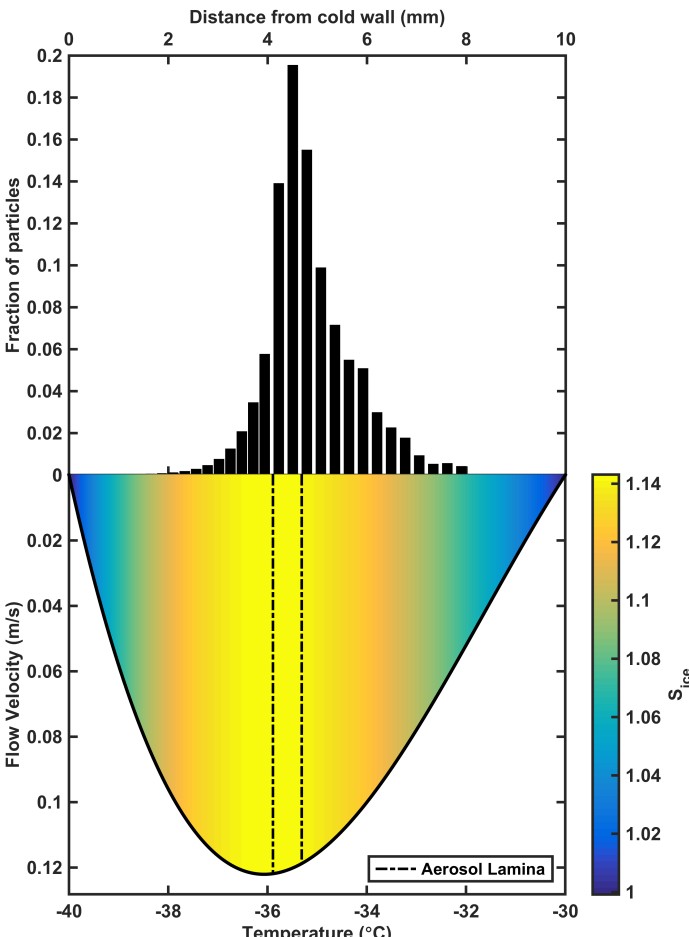

Figure 3. Measured particle distribution across the chamber in the SPIN CFDC (Top; see text for
details) corresponding to the velocity profile and $S_{ice}$ as a function of temperature across the
chamber (Bottom). The dash-dot lines show the location of particles if they were constrained to
the theoretical aerosol lamina. Note that while the peak particle concentration correctly occurs
within the lamina, some particles have migrated into the sheath and are therefore exposed to a
supersaturation significantly lower than the maximum.



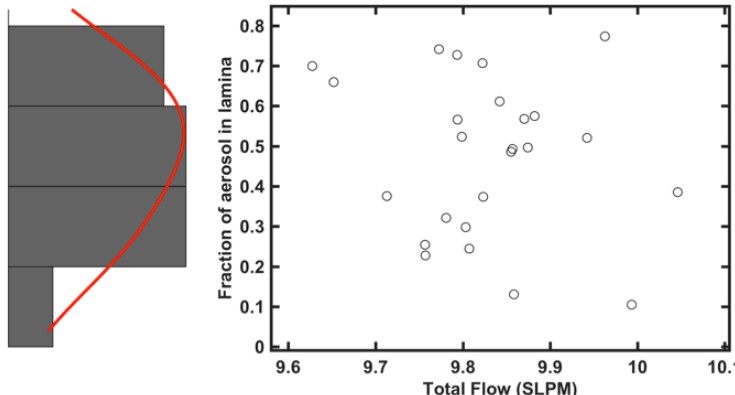



Figure 4. Measured $f_{lam}$ as a function of total flow. In the ideal case, where all particles are
constrained with the dash-dot lines in Figure 2, data points should form a horizontal line at 1.0.
The histogram on the left is a distribution of $f_{lam}$ with the corresponding kernel density estimate
shown in red.





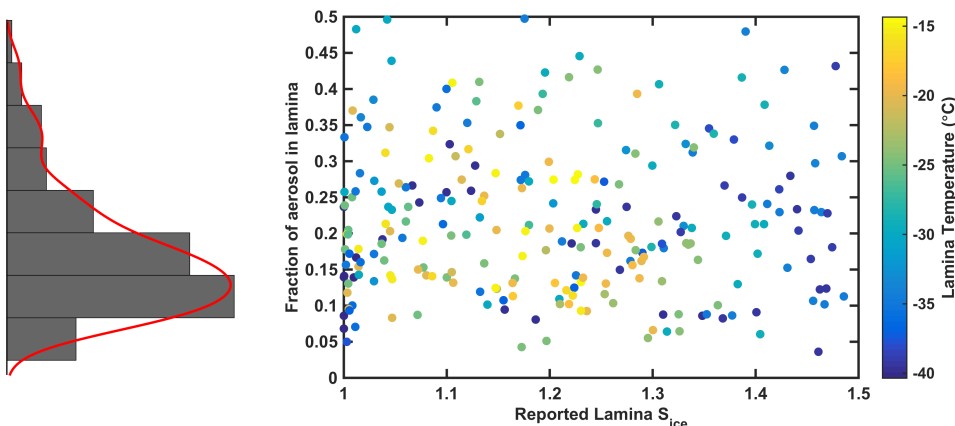

2    Figure 5. Measured $f_{lam}$ as a function of $S_{ice}$ in the aerosol lamina. Temperature for each data

3    point is noted by the color bar. The histogram on the left is the distribution of $f_{lam}$ from the

4    measurements with the corresponding kernel density estimate shown in red.





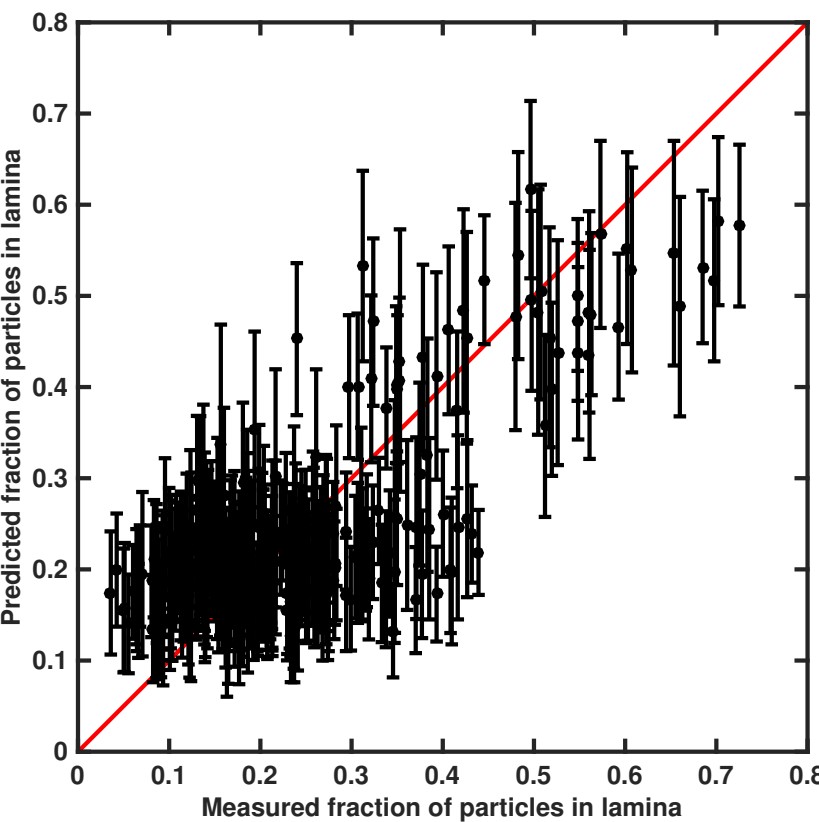

Figure 6. Random forest regression prediction versus measured $f_{lam}$ using the 65 SPIN variables
determined by the algorithm to be most important (see text for details). Data points are the mean
value predicted and error bars correspond to the standard deviation of the predictions by the
random forest. A one-to-one line is shown in red.





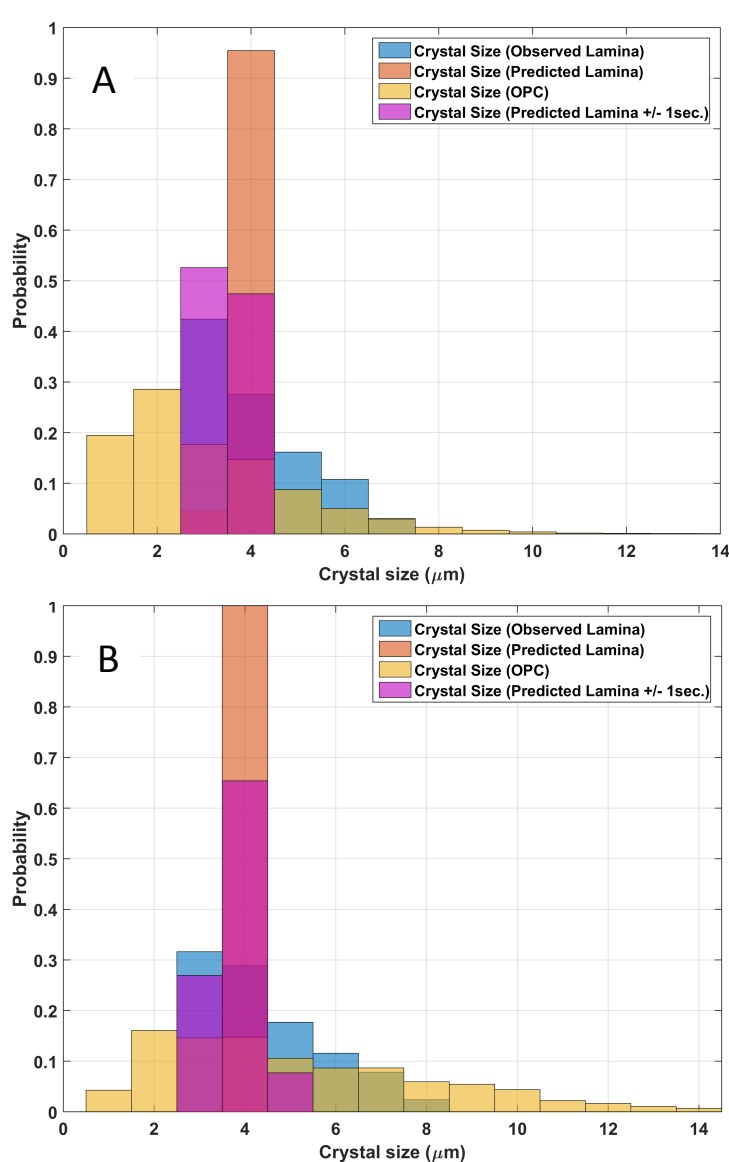

Figure 7. Probability histogram of ice crystals diameter. Both experiments were conducted at -
40° C with Panel A at 102% RH and Panel B at 110% RH, both with respect to liquid water. In
the predicated case all particles are assumed to remain within the lamina, nucleate ice and grow
to the same final size (orange). The crystal size distribution becomes broader if particles are



allowed to exist in the chamber for 1 second shorter or longer than predicted (magenta). Crystal
size predictions using the spreading time indicated in Figure 2 results in the blue histogram. Note
that the crystal size predictions using the spreading time most closely matches the measured
distribution (yellow) from the ZINC chamber.





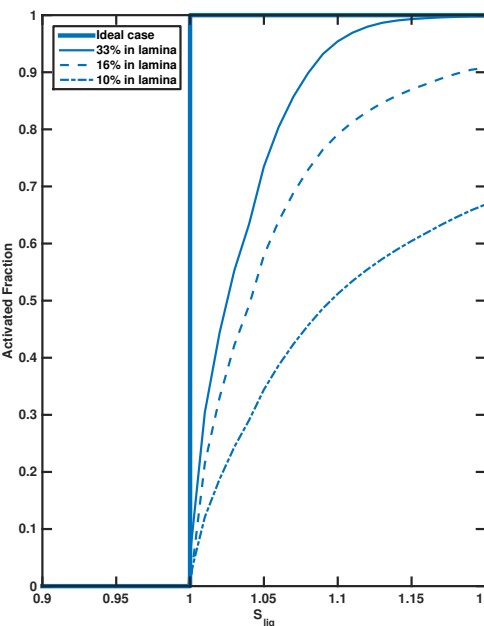

Figure 8. Fraction of particles that activate as a function of saturation with respect to liquid
water. All particles are assumed to be perfect immersion INPs which activate as ice crystals
when exposed to water saturation ($S_{liq}$=1). In the ideal case, where all particles are constrained
within the lamina ($f_{lam}$ 100%; all particles exist within the dash-dot lines in Figure 2), all
nucleation occurs at a CFDC saturation of $S_{liq}$=1 (bold solid line). The other three curves
correspond to increasingly less ideal behavior (i.e., increasingly fewer particles in the lamina and
existing farther from the centerline), corresponding to $f_{lam}$ falling from 33 to 10%. In these cases
an increasingly higher maximum CFDC saturation is required so that the particles farthest from
the centerline experience $S_{liq}$=1.





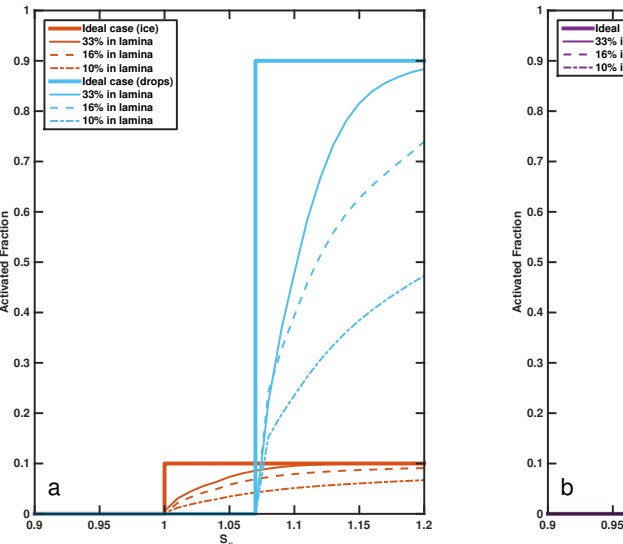
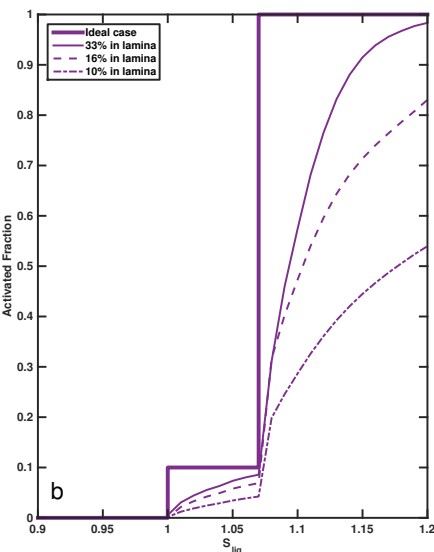

Figure 9. Fraction of particles that activate as a function of saturation with respect to liquid
water. Unlike Figure 6, where all particles are assumed to be perfect immersion INPs which
activate as ice crystals when exposed to water saturation ($S_{liq}$=1), only 10% of particles are
assumed to be perfect immersion INPs. In the ideal case where all particles are constrained
within the lamina ($f_{lam}$ 100%; all particles exist within the dash-dot lines in Figure 2), 10% of
particles nucleate ice at a CFDC saturation of $S_{liq}$=1 (panel a, bold red line). The remaining 90%
of particles are assumed to be perfect CCN that activate at $S_{liq} = 1$. Droplets only survive the
evaporation region of the chamber at $S_{liq} > 1.07$ (corresponding to the SPIN "droplet
breakthrough" point; bold blue line, see text and Garimella et al. (2016) for details). The other
three curves correspond to the increasingly less ideal behavior presented in Figure 6, with $f_{lam}$
falling from 33 to 10%. In these cases an increasingly higher maximum CFDC saturation is
required so that the particles farthest from the centerline experience $S_{liq}$=1 and 1.07. The
resulting activation curves if droplets and ice crystals are indistinguishable (i.e., a composite of
the black and blue traces in panel a) is shown in panel b.