# Peer review of "Uncertainty in counting ice nucleating particles with continuous"

_Atmospheric Chemistry and Physics, 2016_

## Referee Comment (RC1) · Anonymous Referee #1 · 20 Mar 2017

**General Comment** Overall this a well written and very useful paper, if a bit finely focused only on the effect of lamina entry and spreading as a source of uncertainty. Nevertheless, the paper does suggest that this is a major factor of uncertainty in INP measurements with such devices. Two things stood out strongly to me as needing some attention. First, no description of the ice growth model was given, including equations and assumptions on condensation coefficient for ice growth, habit, or at what size particles were started for growth. Second, no demonstration data is given to provide confirmation the errors predicted by the particle transfer studies using ice nucleation data collected in the chambers. Are there any conditions for which higher RH operation is possible in order to confirm that activation appears as predicted in the SPIN instruments (i.e., data consistent with a cf of 2.6-9.5 times)? Past data seems to exist for ZINC in workshops. Can any of these data be utilized? This might make a nice

addition, although I realize that determining a reference RH value could be problematic. Showing an RH scan could at least be useful.

Minor revisions are recommended in general. Specific questions/comments for potentially addressing are listed below.

**Specific Comments**

*Abstract*

Line 13: suggest "instrument theory of operation"

Line 14-16: The sentence ends oddly following the "and". This is a product of the machine learning or the combination of flows? Should there be separate sentences here?

*Introduction*

Page 3, line 20 – Concerning the need for accurate and unbiased measurements, the need exists, but it may or may not be achievable. Why not frame it as "assessing the accuracy and bias of such measurements"?

Page 5, line 3: The referenced study of DeMott et al. (2015) did not only discuss the effect of aerosol spreading outside the lamina, so perhaps this should read "discussed the effect of aerosol "spreading" outside the lamina and other possible factors that together result in a low bias…". This is the first mention of the hypothesis that it is the lamina spreading that is primarily responsible or most important.

*Methodology*

Page 6, lines 6-8: Is there a reason that this ZINC test is done at such a low temperature below that of the mixed phase cloud regime where one thinks of the need to exceed water saturation. Was the thinking just to cover a broad range?

Page 6, lines 14-17: Is there any reason to test different sizes of particles for transfer?

[Figure]

For example, would there be any difference expected for 100 nm versus 1000 nm for whatever phenomenon is responsible for spreading outside of the lamina or whatever leads to undercounting?

Page 6, lines 18-19: In creating a lamina profile of particle arrival times, it is not clear how you decide the position of a particle coming from either side of the central lamina.

Page 7, lines 20-21: What thermodynamic variables besides wall temperatures are meant? What other thermodynamic variables are available in the upper part of the chamber. Why not provide a list of all "features" in a supplemental table?

Page 7, line 22: Where the lamina is "initially" encased within the sheath flows?

*Results and Discussion*

Page 8, line 3: Both walls at $-20\,^{\circ}\mathrm{C}$ and with what uncertainty?

Page 8, line 7: Ice saturation ratio calculated in this case for average wall temperatures?

Page 8, line 18-19: The non-ideality in DeMott et al. was reported for a specific aerosol. Issues of response of aerosol to the chamber conditions is not discussed herein, and one can imagine that it is different for Snomax than for dust particles. Additionally, it could depend on particle hygroscopicity and so forth. Hence, that work was not an expansion on the work of Tobo et al. It regarded a different aerosol entirely, and a point made was that it was not clear if corrections translated to any aerosol. Since this is what is promoted in the present paper, you may wish to make this point.

Page 9, line5: You could perhaps say more here. For example, that the cf could depend on instrument control factors, so would require RFR analysis for any given CFDC.

Page 9, lines 7-9: Since homogeneous freezing is a clear rate process, you would expect a distribution of ice crystal sizes even in that case, no? This could depend on the exact temperature and the nucleation rate at that temperature. It will likely

be monomodal, but I would not expect it to be monodisperse unless you can state the temperature and nucleation rate and show that all particles would be expected to freeze within a very short time after entry into the chamber. I think this paragraph needs some rewriting.

Page 9, lines 16-18: As above, would particles freeze instantly? Even considering their adjustment time in RH and T to chamber conditions that must amount to a second or more?

Page 9, line 23: It is not intuitive in consideration of expected ice crystal growth rates that 4 micron ice particles are expected to be the product at -40 °C. Is it due to the high supersaturation? What is the lamina residence time in ZINC? How large are particles assumed when they freeze? What is the condensation coefficient? What growth rate equation is used? Are spherical ice crystals assumed? There is much missing here in order to evaluate the statement made.

Page 10, lines 8-9: See questions above. How could one possibly know so well the sizes of ice crystals expected if you do not know the deposition coefficient for ice well enough to know what to specify? Are edge effects in the OPC the only explanation of ice crystals less than 3 microns in size? That size could represent a many second or more growth time at -40 °C. And one must assume spherical ice crystals to claim to know the sizes very well.

Page 10, line 14: Does a perfect immersion freezing nuclei imply that they are fully dilute at a condition of a water activity of 1? Most CCN are not activated at 100 percent RH, but immersion freezing can happen before activation if the temperature is cold enough. I think it may have to be stated as a highly idealized and possibly unrealistic example.

Page 11, lines 2-3: A number of 10 percent of aerosols freezing in ambient air is taken as realistic for ambient atmospheric conditions? Under what conditions? Again, I think one can say this is for demonstration purposes, not meant to simulate a real case

except one that might be found in a laboratory or at very low temperatures such as for cirrus parcels.

Page 11, line 12: "of droplets"

Page 11, line 16: perhaps "we propose due to the primary importance of the particle spreading effect." Otherwise you are interpreting another study that discussed multiple processes potentially at play.

*Conclusions*

Page 11: A general comment noted in the above summary comments. It was surprising that no actual ice nucleation data are shown in this paper to support that the spreading effect is realized in the same manner as reported by DeMott et al. (2015).

Page 12: Regarding the comment about instrument geometry, you may wish to say what you mean. For example, some instruments are parallel plate, with lamina edges, while others are cylindrical in design.

Page 12, lines 4-6: There was little exposition given to the idea that small scale flow features at the point of aerosol injection and sheathing are responsible for the observed spreading of particles outside of the lamina. Fluid dynamics simulations might be advised in the future. Perhaps you should say that the only reason for non-ideality explored here was related to sample injection methods. A full analysis might also include particle compositional variability as well, since many INPs are thought to be relatively hydrophobic. One might also ask how the noted behaviors impact "deposition" nucleation?

Page 12, lines 11-14 and beyond: I suggest some revision to the statement here. CFDCs have needed to be operated at higher RH than expected values to inspect immersion freezing. However, an RH value of say 102 percent is not non-physical for Cu clouds, and 106 percent may not be either in wave clouds or very strong elevated convection. Of course, this discussion might be easily resolved by saying the supersaturations are higher than expected for immersion freezing of most particles, rather than stating realism for the atmosphere. Furthermore, S-liq = 1 is not the threshold for immersion freezing. It can be higher or lower in dependence on particle hygroscopic properties, particle size, and temperature, and data in the literature demonstrate this.

Page 12, line 13: INP number concentrations.

Table 1. Please explain some terms better. For example:

1) Lamina saturation at TC3 means using the temperature difference across walls at this elevation to calculate the saturation profile there? Likewise TC4, etc...? 2) What does total volume flow at mass flow controllers mean? Where else do you know it? 3) Probably need to explain the heater concept, since some CFDCs do not heat their coolant.

Figure 1. If you were to draw the sample to scale, would it represent such a small fraction of the flow cross-section?

Figure 3 caption: Don't particles also migrate to higher and lower temperatures?

Figure 4 and caption. Why such a narrow range of total flow, or if constant, why does it vary so much? Why would that be important and why would it even vary? There is a need to state conditions for which these data are collected, that it is for 1-sec pulses, etc...

Figure 9 caption: S-liq > 1.07 is the droplet breakthrough point for what SPIN temperature, or is it uniform? Also, I do not really get what is shown in panel b as a "composite of black and blue traces" from panel a. Why not just say that what is observed by an OPC is shown in panel b?

---

## Referee Comment (RC2) · Anonymous Referee #2 · 11 Apr 2017

The paper describes an important study of the flow conditions inside a CFDC chamber for INP measurements. The study generally confirms the findings presented by DeMott et al., 2015, that a considerable correction factor *cf* needs to be applied in order to correct the CFDC chamber INP measurements. The measurements underestimate INP concentrations apparently because a fraction of the aerosol flow spreads away from the center where the supersaturation is highest. While DeMott et al. suggested a constant factor cf = 3 to be used for their CFDC chamber, the present study for the SPIN instrument finds cf to be variable, ranging between 3 and 10, with a mean value of 4. A Random Forest Regression (RFR) reveals that prediction of the highly variable *cf* can be improved when considering 65 housekeeping variables of the SPIN instrument that seem to be most important in influencing the spreading. Nevertheless, a clear correlation with specific instrumental conditions has not been identified. Therefore, individual INP measurements remain highly uncertain.
The simulated Fig 9b does resemble Fig 2 from DeMott et al, 2015, suggesting that the spreading effect is indeed responsible for the experimentally observed activation curves.

While the subject of this study is highly important for the growing community of researchers using CFDCs for INP measurements, a number of improvements are necessary before the paper can be accepted for ACP.

**General comments**

1) The introduction of the correction factor cf for the SPIN CFDC measurements is important for the experimental determination of INP concentrations. It should be mentioned already in the abstract that a mean correction factor of ~4 is determined for this SPIN instrument, and that the correction factor is highly variable between 3 and 10. The large uncertainty of individual INP measurements due to the large uncertainty of cf should be mentioned in the abstract and discussed in detail in section 3.

2) The measurements concern immersion freezing experiments but the spreading is likely to be present for deposition freezing experiments as well. Please discuss the influence on INP measurements in the deposition freezing mode. Should the same correction factors be applied?

3) While the paper is clearly written in most parts, some parts are imprecise and not well-written. This concerns especially the first half of section 3 (pages 8 and 9, Table 1 and Figs 2-5, see specific comments below). The descriptions are not detailed and accurate enough and for several paragraphs it is difficult to extract the main message the authors want to convey. For example, the description and interpretation of Fig. 4 is only 2.5 lines (p 8, l 3-5), and the exact purpose of this Figure does not become clear to the reader. Is it supposed to show that $f_{lam}$ changes more or less erratically between 0.1 and 0.8 for conditions that are kept as constant as possible? What does it tell about the reproducibility and uncertainty of the INP measurements?

4) Can effects of thermophoresis be excluded? Do the aerosol particles potentially leave the theoretical aerosol lamina due to thermophoresis?

5) A particle that moves slightly outside the central lamina but still in the yellow region of Fig 3 should still be activated and growing efficiently. Is the assumption correct that all particles that leave the central lamina once (and are therefore counted in the "late" tail of the pulse) are not activated and cannot be measured as INP (therefore necessitating the large correction factors)?

6) The most likely reasons for the observed spreading effect and for the discrepancy between the ideal instrument and the real measurements should be discussed. Are uncontrolled eddy turbulences the main/only reason for the spreading?

**Specific comments**

1.) The manuscript switches frequently between the ZINC and the SPIN instrument and sometimes it is unclear which specific instrument is meant (e.g. Fig 2: pulses are shown for ZINC, Fig 3: SPIN results, Fig 4 which instrument? SPIN? (please include instrument name in Figure caption), Fig 5 SPIN, etc.).

2.) p 5, l 20-21: Why did you use 1-second pulses for SPIN and 10 sec pulses for ZINC?
Did you measure the $CPC_{in}$ pulse every time and are the blue and red trace in Fig 2 measured for the identical pulse? The blue pulse in panel A seems to be shorter than 10 seconds. How long is the transfer time through the SPIN and ZINC chambers?
The example of Fig 2 does not seem to be a typical one: with $f_{lam}$ = 77.7 and 76.2 % it is much higher than all the values displayed in Figs 4 and 5. According to Fig 5, the most frequent $f_{lam}$ is in the range of 10-15%; and the average $f_{lam}$ is argued to be ~25% (see comment to Fig 5 below). Please display (also) the measured $CN_{out}$ for such a more typical case. Does it make sense to present the percentages for $f_{lam}$ with a decimal place?

3.) p. 6, line 13-19 and Figure 3: the description is not sufficient. In the Figure caption it is stated that the particle distribution is "measured across the chamber". Is this true? In the text of p 6 it says that "combining the arrival pulse with the shape of the velocity profile the corresponding distribution of particles across the width of the chamber can be determined". How is this distribution determined in detail? This seems to be a complicated matter to me that would require CFD modelling, etc.? Do you derive a different distribution for each measurement pulse?
The term "measured across the chamber" would indicate that CN measurements are made at the end of the chamber at different distances from the cold wall. Please use such a term only if such measurements were actually performed.

4.) Figure 4:
The y-axis should range from 0 to 1. There are only 25 data points shown, the text talks about 30 data points. Are all data points displayed?
It is stated that the tests shown in Fig 2 were done at +20°C (p8, l 3 and p6, l 14). Does this mean that there was no cooling applied and the chamber walls were at room temperature for these measurements? Are these conditions transferable to realistic flow conditions? It would indeed be interesting to see in how far the pulses change between a warm chamber at constant temperature and a chamber operating with the two different cold wall temperatures.

5.) Figure 5:
There seems to be an error in the Figure: The y-scale of the panel on the right does not correspond with the histogram on the left. The text claims that the mean of the distribution of $f_{lam}$ is at 0.25, in the graph on the left the mean seems to be around 0.15.

6.) p 8, l 10: Here it is reported that values for $f_{lam}$ range from 3 to 73 %. Why are values of 76 and 77 % reported in Fig 2? (difference between ZINC and SPIN?)

7.) Figure 7:
The colors are hard to discern. Four colors are shown in the legend, but several other colors are shown in the graph. This is potentially because overlapping colors result in "new" colors? A different representation would be helpful, e.g. show the four probability distributions not as histograms but as line plots.

8.) p 9-p 10: The ice growth model is insufficiently described. What are the assumptions? How are things calculated? I agree with the other reviewer that this needs considerably more discussion.

9.) p 10, l 13: $f_{lam}$ and cf are not equivalent, but rather "inversely equivalent".

**Technical comments**

p 7, l 15: "import" → "important"

p 7, l 21: "middle-top of the SPIN" → "middle and top section of the SPIN"

p 11, l 12: "activation curves of droplets… "

p 11, l 13: "black and blue" → "red and blue".

p 11, l 18: "4. Conclusions"  (section numbering)

p12, l 17: "variability to be conducted"

p 27, l 3: "Unlike Figure 6" should read "Unlike Figure 8"

p 27, l 15 : "black and blue" → "red and blue"

---

## Author Comment (AC1) · 19 May 2017

We would like to thank both Reviewers for their careful reading and thoughtful comments on our manuscript. We have made the changes they suggested and provide a point by point response below with the comment directly followed by the response. We believe this is a much improved paper as a result.

Reviewer 1 (Major points extracted from first paragraph with overview text omitted for clarity; we quote these points here and refer the Reviewer / Editor to the comment in the main body of the review where they apply.)

Major Points

1. First, no description of the ice growth model was given, including equations and

assumptions on condensation coefficient for ice growth, habit, or at what size particles were started for growth.

This point is repeated and addressed in response to that for Page 9, line 23.

2. Second, no demonstration data is given to provide confirmation the errors predicted by the particle transfer studies using ice nucleation data collected in the chambers. Are there any conditions for which higher RH operation is possible in order to confirm that activation appears as predicted in the SPIN instruments (i.e., data consistent with a cf of 2.6-9.5 times)? Past data seems to exist for ZINC in workshops. Can any of these data be utilized?

This point is repeated and addressed in response to the first point in the Conclusions.

Specific Comments

Abstract Line 13: suggest "instrument theory of operation"

Change made.

Line 14-16: The sentence ends oddly following the "and". This is a product of the machine learning or the combination of flows? Should there be separate sentences here?

Sentence broken at "and" with wording changed to "We use a machine learning approach to show that non-ideality is most likely due to small scale flow features where the aerosols are combined with sheath flows. Machine learning is also used to minimize the uncertainty in measured INP concentrations. We suggest that detailed measurement, on an instrument-by-instrument basis, be performed to characterize this uncertainty."

Introduction Page 3, line 20 – Concerning the need for accurate and unbiased measurements, the need exists, but it may or may not be achievable. Why not frame it as "assessing the accuracy and bias of such measurements"?

Change made.

Page 5, line 3: The referenced study of DeMott et al. (2015) did not only discuss the effect of aerosol spreading outside the lamina, so perhaps this should read "discussed the effect of aerosol "spreading" outside the lamina and other possible factors that together result in a low bias: : :". This is the first mention of the hypothesis that it is the lamina spreading that is primarily responsible or most important.

We agree with the reviewer; we have concentrated on the spreading issue in this paper but did not intend to ignore the other possible factors outlined by DeMott et al. We have made the suggested change to "DeMott et al. (2015) discussed the effect of aerosol "spreading" outside the lamina as well as other possible factors that, in combination, could contribute to a low bias in the number of INP measured."

Methodology Page 6, lines 6-8: Is there a reason that this ZINC test is done at such a low temperature below that of the mixed phase cloud regime where one thinks of the need to exceed water saturation. Was the thinking just to cover a broad range?

To demonstrate our thinking we have added "This temperature was used to determine effect of temperature on particle loss from the lamina and because it is in the homogeneous freezing regime where all test particles are able to nucleate ice. At this lamina temperature the difference between the wall temperatures is larger for a given supersaturation than at a higher temperature and this maximized any resulting turbulence effect on particle migration from the lamina."

Page 6, lines 14-17: Is there any reason to test different sizes of particles for transfer? For example, would there be any difference expected for 100 nm versus 1000 nm for whatever phenomenon is responsible for spreading outside of the lamina or whatever leads to undercounting?

We did not observe differences from 100 to 200 nm and could not definitively attribute effects when using polydisperse aerosol in the field. We do not believe diffusive or

other effects are equivalent to the mixing/turbulence effects we have qualified this at the end of this section with "We note that diffusive and other forces may differ across particle sizes. Monodisperse particles were tested under laboratory and polydisperse particles under field conditions; future studies may consider a full range of particle sizes applicable to particular CFDCs."

Page 6, lines 18-19: In creating a lamina profile of particle arrival times, it is not clear how you decide the position of a particle coming from either side of the central lamina.

The text has been expanded: "Combining information from a measured particle pulse and a calculated velocity profile, the corresponding distribution of particles across the width of the chamber for that particular pulse can be inferred (Figure 3). Buoyancy effects on mean chamber flow and mean particle position are accounted for in the calculation of the velocity profiles (Rogers, 1988). The particle distributions are reconstructed by assigning the first detected particles to the maximum velocity position in the calculated flow profile and the assigning peak particle concentration to the calculated lamina position (about which spreading occurs). The particles in the tail of the pulse are assigned positions corresponding to their relative velocities, which are derived from their relative arrival times."

Page 7, lines 20-21: What thermodynamic variables besides wall temperatures are meant? What other thermodynamic variables are available in the upper part of the chamber.

Replaced with "saturation conditions" (see also Table 1.)

Why not provide a list of all "features" in a supplemental table?

As stated in the text, an exponential fall of importance was observed and we present those within the first two e-foldings (see paragraph at end of Section 2 and Table 1). Presenting a full list of housekeeping variables beyond those (and used in the reduced model) has no bearing on the results and we believe it could actually create confusion

for the reader.

Page 7, line 22: Where the lamina is "initially" encased within the sheath flows? Results and Discussion

"initially" added as a qualifier

Page 8, line 3: Both walls at -20 C and with what uncertainty?

There is a misunderstanding here; there is no "-" in the text. These were isothermal experiments at 20° C where this was performed intentionally so there would be no temperature difference. This point is relevant for latter experiments so we have added "Wall temperatures remained within $\pm$ 1 degree of the set point for all experiments."

Page 8, line 7: Ice saturation ratio calculated in this case for average wall temperatures?

Clarified as "In Figure 5, flam is plotted against the lamina temperature and ice saturation ratio (Sice) calculated for walls at ice saturation and with temperatures corresponding to the measurement average value (Garimella et al., 2016)."

Page 8, line 18-19: The non-ideality in DeMott et al. was reported for a specific aerosol. Issues of response of aerosol to the chamber conditions is not discussed herein, and one can imagine that it is different for Snomax than for dust particles. Additionally, it could depend on particle hygroscopicity and so forth. Hence, that work was not an expansion on the work of Tobo et al. It regarded a different aerosol entirely, and a point made was that it was not clear if corrections translated to any aerosol. Since this is what is promoted in the present paper, you may wish to make this point.

We agree this statement was overly simplistic and have expanded this to "DeMott et al. (2015) noted the non-ideality, including due to particles spreading beyond the lamina, in the Colorado State University CFDC chamber. They proposed the use of a "calibration factor" (cf) = 3 by which the measured INP number could be multiplied to provide a corrected value. Previous studies, including Tobo et al. (2013), used cf = 1; this

corresponds to an assumption that all particles in a CFDC exist within the lamina. It should be noted particle properties, such as size, shape and hygroscopicity may have an effect on the correction factor and that the value found by DeMott et al. (2015) may not be universal even for that CFDC. The value of cf = 3 does correspond to a constant 33% of particles existing within the lamina, regardless of flow or thermodynamic state, for those experiments." which we believe is in keeping with the reviewer's suggestion here.

Page 9, line5: You could perhaps say more here. For example, that the cf could depend on instrument control factors, so would require RFR analysis for any given CFDC.

Suggested change made to "This suggests that cf could depend on various instrument control factors and would require RFR analysis for the various CFDC configurations."

Page 9, lines 7-9: Since homogeneous freezing is a clear rate process, you would expect a distribution of ice crystal sizes even in that case, no? This could depend on the exact temperature and the nucleation rate at that temperature. It will likely be monomodal, but I would not expect it to be monodisperse unless you can state the temperature and nucleation rate and show that all particles would be expected to freeze within a very short time after entry into the chamber. I think this paragraph needs some rewriting.

We start this response point by noting that this and the following four points all request an expansion of the ice growth calculation and the assumptions therein. We have expanded this section by several paragraphs and believe that this presentation is much more clear for the reader as a result. We are also compelled to point out that although the Reviewer initially raises important questions that were in need of clarification, the final points, especially that at "Page 10, lines 8-9", demonstrates a misunderstanding: In the initial manuscript we started this section with the qualification that this was a supporting argument, not our main point, and included multiple qualifying statements throughout. This remains true in the revised manuscript. It is not correct for the Reviewer to argue that we are trying to make points by assuming we know crystal sizes with high certainty; that is not and never was a central argument of this manuscript. We hope the review can proceed with this as context.

Regarding this specific point, we agree with the reviewer that the crystal size distribution should be quasi-monodisperse and monomondal since there is a variation in temperature within the aerosol sample layer even under ideal CFDC operation, which also leads to a variation in RHi. However, the particles should equilibrate with the internal chamber conditions within approximately the first second of entering the chamber and then freeze almost instantly due to the very high nucleation rate at these conditions. To clarify this in the text we have reworded this paragraph at the reviewer's suggestion to "Further evidence to support the spreading effect is provided by the size of the ice crystals measured at the output of a CFDC. Theoretically, a monodisperse population of an aerosol composition that only nucleates ice homogeneously should exhibit freezing almost the same time and location within a CFDC chamber. This is because particles should equilibrate with the internal chamber conditions within ∼1 second of entering the chamber and then freeze rapidly due to the resulting nucleation rate at these conditions (Koop et al., 2000). This should translate to a quasi-monodisperse ice crystal size distribution at the chamber output; the size of the crystals should be a function of the chamber RH and temperature and equivalent to the amount of vapor-deposited water under these conditions. Therefore, differences in crystal size should primarily be due to particles that leave the lamina and experience varying supersaturations and residence times in the chamber."

Page 9, lines 16-18: As above, would particles freeze instantly? Even considering their adjustment time in RH and T to chamber conditions that must amount to a second or more?

We agree, the time scale would be on the order of a second. We have added removed "immediately" from the text and added the clarification "In order to consider the effect spreading beyond the lamina, aerosol particles were assumed to nucleate ice upon

entering the chamber since the -40° C lamina temperature was below that required for homogeneous ice nucleation. We note there is a short delay not directly accounted for in this calculation for thermal equilibration ($\sim$ 0.1 – 0.8 seconds at -40 °C assuming heat transfer coefficients for the NH4NO3 particles) and $\sim$1.0 seconds for the particles to travel to the ice coated region of the chamber (Stetzer et al., 2008). The combination of velocity profile and residence time from the pulse experiments (Figure 2) were then used to determine the location of the particles in the lamina and therefore the time they were exposed to variable supersaturation and the subsequent size to which they would grow."

Page 9, line 23: It is not intuitive in consideration of expected ice crystal growth rates that 4 micron ice particles are expected to be the product at -40 C. Is it due to the high supersaturation? What is the lamina residence time in ZINC? How large are particles assumed when they freeze? What is the condensation coefficient? What growth rate equation is used? Are spherical ice crystals assumed? There is much missing here in order to evaluate the statement made.

Note that this response also answers the first major point made in the initial review paragraph. In response we now include more information at this location as to the ice growth model : "The baseline crystal size was when all particles remained within the predicated lamina (i.e., within the dash lines in Figure 1). Ice crystal size was calculated per the formulation of Rogers and Yau, 1989). Crystals were assumed to be spherical due to the ice forming homogeneously from sub-micrometer diameter particles (Järvinen et al., 2016). The initial ice crystal size was assumed to be that of ammonium nitrate at the initial dry diameter, which may be a slight underestimation due to hygroscopic growth before freezing occurred. However, calculations with a doubling of the initial particle size had minimal impact on the final crystal size so this assumption was maintained for all calculations. An accommodation coefficient of 0.2 (Strotski et al., 2013) and the calculated residence time in the lamina ($\sim$10 seconds) were used to predict the crystal sizes. The calculations resulted in monodisperse ice crystals at $\sim$4

micrometers diameter (orange histogram)."

Page 10, lines 8-9: See questions above. How could one possibly know so well the sizes of ice crystals expected if you do not know the deposition coefficient for ice well enough to know what to specify? Are edge effects in the OPC the only explanation of ice crystals less than 3 microns in size? That size could represent a many second or more growth time at -40 C. And one must assume spherical ice crystals to claim to know the sizes very well.

We understand that reviewer has concerns about the ice growth model and we have – in addition to the changes outlined in the last four points – also added "It is important to note that the previous calculations are from a simple ice growth model and are used to illustrate that observed crystal size distributions are also consistent with particles spreading beyond the lamina." in this section. We request the Reviewer keep this in mind as we were not making statements or conclusions based on the level of certainty suggested in their point.

That said, we include the further qualifying statement regarding smaller crystals per the reviewer's comments as "In addition to spreading, there are other reasons that crystals smaller than the theoretical size might exist. These include uncertainty in shape, refractive index and crystals that are undersized by passing through the edge of the ZINC OPC (Stetzer et al., 2008). Furthermore, literature values of accommodation coefficient ranges between 0.2 and 1 (e.g.,Skrotzki et al., 2013). Here, 0.2 was used for these calculations. A value of 0.1 would result in $\sim$10% smaller crystals, which is still not sufficient to fully account for the smaller crystals < 2 micrometers diameter observed in the OPC."

Page 10, line 14: Does a perfect immersion freezing nuclei imply that they are fully dilute at a condition of a water activity of 1? Most CCN are not activated at 100 percent RH, but immersion freezing can happen before activation if the temperature is cold enough. I think it may have to be stated as a highly idealized and possibly unrealistic

example.

We agree with the reviewer and this was the reason we described Figure 8 as "idealized" and then used the term "perfect" (quotes here in original text) and called out the assumption they activated at exactly 100% in the initial manuscript. We have now further qualified this as "The aerosol population is assumed to be idealized "perfect" immersion mode INP that form ice crystals immediately upon exposure at water saturation (Sliq=1)"

Page 11, lines 2-3: A number of 10 percent of aerosols freezing in ambient air is taken as realistic for ambient atmospheric conditions? Under what conditions? Again, I think one can say this is for demonstration purposes, not meant to simulate a real case except one that might be found in a laboratory or at very low temperatures such as for cirrus parcels.

As with the preceding point, we believe the reviewer might have missed the qualifications that these plots were meant to be idealized representations, not atmospheric examples, in the original text. To make this more clear we have further qualified this line with "Figure 9 expands on Figure 8 by considering another idealized case, but one more applicable to measurement of an ambient aerosol population. In this case only 10% of the particles are perfect immersion mode INP, whereas the rest are cloud condensation nuclei (CCN) that activate at exactly Sliq=1."

Page 11, line 12: "of droplets"

Corrected

Page 11, line 16: perhaps "we propose due to the primary importance of the particle spreading effect." Otherwise you are interpreting another study that discussed multiple processes potentially at play.

Suggested wording change made.

Conclusions

Page 11: A general comment noted in the above summary comments. It was surprising that no actual ice nucleation data are shown in this paper to support that the spreading effect is realized in the same manner as reported by DeMott et al. (2015).

Regarding ZINC data from previous workshops (ICIS 2007, DeMott et al., 2011) we prefer not to use these data as the experiments were performed with a goal to observe only the onset RH of activation. In addition, ZINC was still under development during the workshop conducted in 2007 as such it may be pre-mature to use those data for a comparison.

However, for this work, we present another experiment with a complete activation spectrum (see Figure 1 below) in ZINC taken at 233 K for homogeneous freezing of 400 nm (dry diameter) NH4NO3 particles. For these conditions, we don't have to compare to another immersion freezing device since the particle freezing efficiency is not of concern given that homogeneous freezing should render all particles frozen for $S_{liq} \geq 0.99$ (Koop et al., 2000)

For approximately these conditions (233 K and $S_{liq}$ 1.02) we have reported that $\sim$ 25% of the particles escape the lamina in ZINC (see caption Figure 2). As is predicted from the curves shown in Figure 8, the deviation from a step function form of the curve is indicative of particles escaping the lamina, thus broadening the $S_{liq}$ range at which complete activation (100% of particles freezing) is observed. In the figure below (new figure in revised version (Figure 9)), the range of $S_{liq}$ that the aerosol particles in the lamina are exposed to is shown by the shaded region. Within this range $\sim$ 70% of the particles freeze homogeneously in ZINC, consistent with the pulse tests in Figure 2 of the manuscript, considering counting uncertainties of $\sim$14% arising from the CPC and OPC. However, in order to observe 100% of particles freezing, one must increase $S_{liq}$ to 1.05 in ZINC suggesting that $\sim$30% of the particles escape the lamina in this case. We have now added this explanation to the revised manuscript on page 13 lines 3-12.

Fraction of 400 nm (dry diameter) NH4NO3 particles freezing homogenously as a function of Sliq in ZINC at -40 °C. According to Koop et al. (2000), 400 nm salt particles at -40 °C should freeze homogenously at Sliq ∼ 0.99. Shaded region indicates the range and uncertainty of Sliq that the aerosol in the lamina are exposed to.

Page 12: Regarding the comment about instrument geometry, you may wish to say what you mean. For example, some instruments are parallel plate, with lamina edges, while others are cylindrical in design.

Now included "(i.e., parallel plate, cylindrical, etc.)"

Page 12, lines 4-6: There was little exposition given to the idea that small scale flow features at the point of aerosol injection and sheathing are responsible for the observed spreading of particles outside of the lamina. Fluid dynamics simulations might be advised in the future.

This was suggested in the initial manuscript and remains in the revised version "...drawing comparisons to computational fluid dynamics simulations to complement the RFR statistical modeling..."

Perhaps you should say that the only reason for non-ideality explored here was related to sample injection methods.

This is incorrect. The RFR points to this being the case since the location of the most important factors were in this area. A different RFR result would have caused us to consider different reasons / chamber features. As such this would not be a correct statement / qualification.

A full analysis might also include particle compositional variability as well, since many INPs are thought to be relatively hydrophobic. One might also ask how the noted behaviors impact "deposition" nucleation?

We agree and have now added "consider other freezing regimes such as depositional nucleation of ice" to the list of future work / suggestions.

Page 12, lines 11-14 and beyond: I suggest some revision to the statement here. CFDCs have needed to be operated at higher RH than expected values to inspect immersion freezing. However, an RH value of say 102 percent is not non-physical for Cu clouds, and 106 percent may not be either in wave clouds or very strong elevated convection. Of course, this discussion might be easily resolved by saying the supersaturations are higher than expected for immersion freezing of most particles, rather than stating realism for the atmosphere. Furthermore, S-liq = 1 is not the threshold for immersion freezing. It can be higher or lower in dependence on particle hygroscopic properties, particle size, and temperature, and data in the literature demonstrate this.

We believe the reviewer might have missed the wording in this section. We have not discussed atmospheric cloud supersaturation. Instead, we made reference to droplet nucleation in e.g. CCN instruments where 1.02 and great are unrealistic supersaturations "By contrast, CCN instruments routinely activate essentially all particles into droplets at 1.01 – 1.02.". This is a valid comparison and it is the one that was stated in the initial, and remains in the revised, manuscript.

Page 12, line 13: INP number concentrations.

Corrected

Table 1. Please explain some terms better. For example: 1) Lamina saturation at TC3 means using the temperature difference across walls at this elevation to calculate the saturation profile there? Likewise TC4, etc...?

Table caption now revised to "List of the ten most important features from the RFR. TC corresponds to Thermocouple and H to heater where numbers correspond to locations described in Garimella et al. (2016). Saturation is calculated at specific locations using temperature measurements and assuming walls exist at ice saturation per the method outlined in Garimella et al. (2016). The features are predominantly located in the top and middle sections of the chamber."

2) What does total volume flow at mass flow controllers mean? Where else do you know it?

This is "total volume flow"; MFC now eliminated for clarity.

3) Probably need to explain the heater concept, since some CFDCs do not heat their coolant.

Within the caption now included "(supplemental heating is used to maintain wall isothermality in the SPIN chamber)"

Figure 1. If you were to draw the sample to scale, would it represent such a small fraction of the flow cross-section?

This figure starts with the statement "Schematic representation of an idealized CFDC" – exact dimensions are not meant to be represented.

Figure 3 caption: Don't particles also migrate to higher and lower temperatures?

Reworded to "some particles have migrated into the sheath and are therefore exposed to higher and lower temperatures and supersaturation lower than the maximum."

Figure 4 and caption. Why such a narrow range of total flow, or if constant, why does it vary so much? Why would that be important and why would it even vary? There is a need to state conditions for which these data are collected, that it is for 1-sec pulses, etc.

Flow was varied slightly around the nominal value "varied around a nominal value of 9.8 slpm." added to text. The pulse length was explicitly stated in the text; we do not believe it should be repeated here.

Figure 9 caption: S-liq > 1.07 is the droplet breakthrough point for what SPIN temperature, or is it uniform? Also, I do not really get what is shown in panel b as a "composite of black and blue traces" from panel a. Why not just say that what is observed by an OPC is shown in panel b?

[Figure]

This has been corrected to "to a "droplet breakthrough" point"; we did not mean to imply a specific instrument or condition. We have also reworded the main text to make this clear. Also corrected (see Reviewer 2) is red, not black. Now further explained with "as is the case for an OPC".

―――――――――――――――――

---

## Author Comment (AC2) · 19 May 2017

We would like to thank both Reviewers for their careful reading and thoughtful comments on our manuscript. We have made the changes they suggested and provide a point by point response below with the comment followed by the response. We believe this is a much improved paper as a result.

Reviewer #2

(overview paragraph omitted for clarity)

General comments

1) The introduction of the correction factor cf for the SPIN CFDC measurements is important for the experimental determination of INP concentrations. It should be mentioned already in the abstract that a mean correction factor of ∼4 is determined for this SPIN instrument, and that the correction factor is highly variable between 3 and 10. The large uncertainty of individual INP measurements due to the large uncertainty of cf should be mentioned in the abstract and discussed in detail in section 3.

We agree with the reviewer and explicitly state "We find here variable correction factors from 1.5-9.5, consistent with previous literature values." in abstract. Not limited by length, we have included additional wording in section 3, per both reviewers suggestions. This is the paragraph starting with "DeMott et al. (2015)..."

2) The measurements concern immersion freezing experiments but the spreading is likely to be present for deposition freezing experiments as well. Please discuss the influence on INP measurements in the deposition freezing mode. Should the same correction factors be applied?

Please note that a similar comment was made by Reviewer 1. We now explicitly call for depositional regime work in the conclusions. In the introduction we add "In this work we specifically considered effects in the regime supersaturated with respect to liquid water (immersion freezing) but believe these results are also applicable in the sub-saturated regime (depositional nucleation) as well."

3) While the paper is clearly written in most parts, some parts are imprecise and not well written. This concerns especially the first half of section 3 (pages 8 and 9, Table 1 and Figs 2‐ 5, see specific comments below). The descriptions are not detailed and accurate enough and for several paragraphs it is difficult to extract the main message the authors want to convey.

In response to this point we have re-read and made changes in these sections. The Table 1 caption was re-written in response to this and a point by Reviewer 1. We believe these are more clear as a result.

For example, the description and interpretation of Fig. 4 is only 2.5 lines (p 8, l 3‐5),

and the exact purpose of this Figure does not become clear to the reader. Is it supposed to show that flam changes more or less erratically between 0.1 and 0.8 for conditions that are kept as constant as possible? What does it tell about the reproducibility and uncertainty of the INP measurements?

We have attempted to improve the clarify of this caption with the addition of "This figure illustrates that the ideal condition is not realized and that even within a few % of the nominal total flow of 9.8 lpm the fraction of particles in the lamina is not predictable."

4) Can effects of thermophoresis be excluded? Do the aerosol particles potentially leave the theoretical aerosol lamina due to thermophoresis?

(re-ordered similar points)

6) The most likely reasons for the observed spreading effect and for the discrepancy between the ideal instrument and the real measurements should be discussed. Are uncontrolled eddy turbulences the main/only reason for the spreading?

The RFR method suggests the location in the chamber where the spreading effects are most highly correlated with. We now expand on this at the end of section 2. While this is consistent with e.g. turbulence in the aerosol injection region we can not rule out thermophoresis. "The reduced RFR subset included 65 variables including wall temperature, flows, and saturation conditions predominantly in the middle and top sections of the SPIN chamber (Garimella et al., 2016); this is the region of the chamber where aerosol is initially encased within the sheath flows. This suggests that turbulence or other small-scale flow features in this region are responsible for the spreading effect in the region where the particle flow is injected into the chamber. However, we can not preclude that other processes taking place in this region, such as thermophoresis, are not also partially responsible."

5) A particle that moves slightly outside the central lamina but still in the yellow region of Fig 3 should still be activated and growing efficiently. Is the assumption correct that all

particles that leave the central lamina once (and are therefore counted in the "late" tail of the pulse) are not activated and cannot be measured as INP (therefore necessitating the large correction factors)?

This point regards two different effects. First is the question if particles leave the lamina in which they are supposed to remain. We show that this is not the case in Figure 3-5 and the associated text. Particles that do leave the lamina may still activate but they will only do so for lamina conditions in excess of the "true" activation value for that particle. This is argued in the revised ice growth model in Section 3 and the idealized figures 8 and 9. In the case of Figures 8 and 9 this point is found in the text in the revised last two paragraphs of Section 3.

6) This point was moved up as it related to point 4)

Specific comments

1.) The manuscript switches frequently between the ZINC and the SPIN instrument and sometimes it is unclear which specific instrument is meant (e.g. Fig 2: pulses are shown for ZINC, Fig 3: SPIN results, Fig 4 which instrument? SPIN? (please include instrument name in Figure caption), Fig 5 SPIN, etc.).

Figure captions with instrumental data now explicitly call out SPIN or ZINC.

2.) p 5, l 20‐21: Why did you use 1‐second pulses for SPIN and 10 sec pulses for ZINC? Did you measure the CPCin pulse every time and are the blue and red trace in Fig 2 measured for the identical pulse? The blue pulse in panel A seems to be shorter than 10 seconds.

This is now clarified with "In the case of SPIN this was a 1 second pulse while for ZINC a ∼10 second pulse was used with automated and manual valves, respectively." The purpose of the pulse tests is to show the difference in arrival time between CPCin and CPCout. "Under ideal conditions, regardless of duration, this should correspond to an equivalent particle pulse at the chamber outlet. "

How long is the transfer time through the SPIN and ZINC chambers?

In both cases this is ~10 seconds, given in the referenced papers that start this section, but not directly applicable here.

The example of Fig 2 does not seem to be a typical one: with flam = 77.7 and 76.2 % it is much higher than all the values displayed in Figs 4 and 5. According to Fig 5, the most frequent flam is in the range of 10‐15%; and the average flam is argued to be ~25% (see comment to Fig 5 below). Please display (also) the measured CNout for such a more typical case.

There is some confusion here, which might have been cleared up with the addition of the instruments to the figure captions. The Reviewer is mixing two instruments and several sets of temperature and conditions. These figures are not meant to be equivalent, they show different regimes. We belie this is now clear with the aforementioned changes.

Does it make sense to present the percentages for flam with a decimal place?

We agree and have revised the precision throughout the manuscript.

3.) p. 6, line 13‐19 and Figure 3: the description is not sufficient. In the Figure caption it is stated that the particle distribution is "measured across the chamber". Is this true? In the text of p 6 it says that "combining the arrival pulse with the shape of the velocity profile the corresponding distribution of particles across the width of the chamber can be determined". How is this distribution determined in detail? This seems to be a complicated matter to me that would require CFD modeling, etc.? Do you derive a different distribution for each measurement pulse? The term "measured across the chamber" would indicate that CN measurements are made at the end of the chamber at different distances from the cold wall. Please use such a term only if such measurements were actually performed.

In this context "across the chamber" is meant to indicate the position with respect to

the lamina, not a direct measurement. Also, we do derive a different distribution for each measurement pulse. To clarify this additional text has been added in this location "Combining information from a measured particle pulse and a calculated velocity profile, the corresponding distribution of particles across the width of the chamber for that particular pulse can be inferred (Figure 3). Buoyancy effects on mean chamber flow and mean particle position are accounted for in the calculation of the velocity profiles (Rogers, 1988). The particle distributions are reconstructed by assigning the first detected particles to the maximum velocity position in the calculated flow profile and the assigning peak particle concentration to the calculated lamina position (about which spreading occurs). The particles in the tail of the pulse are assigned positions corresponding to their relative velocities, which are derived from their relative arrival times. Since particles in the arrival tail could fall on either side of position distribution, this ratio is assigned based on matching the exponential fit of the unambiguous portion of the data (between the initial particle arrival and the peak concentration)."

4.) Figure 4: The y‐axis should range from 0 to 1. There are only 25 data points shown, the text talks about 30 data points. Are all data points displayed? It is stated that the tests shown in Fig 2 were done at +20°C (p8, l 3 and p6, l 14). Does this mean that there was no cooling applied and the chamber walls were at room temperature for these measurements? Are these conditions transferable to realistic flow conditions? It would indeed be interesting to see in how far the pulses change between a warm chamber at constant temperature and a chamber operating with the two different cold wall temperatures.

The y-axis of Figure 4 now has a range of 0-1. The text and figure also now both reflect the correct number of data points used in the figure (25). There was no cooling applied and the chamber walls for the tests in Figure 4, and they were at room temperature for these measurements. These conditions were chosen to examine particle spreading with the fewer variables (e.g. thermophoresis in the chamber, increased temperature variability, etc.) than with cooled walls. For comparison, Figure 4 shows data from

a warm chamber at constant temperature, and Figure 5 shows data from a cooled chamber operating with various temperature gradients.

5.) Figure 5: There seems to be an error in the Figure: The y‐scale of the panel on the right does not correspond with the histogram on the left. The text claims that the mean of the distribution of flam is at 0.25, in the graph on the left the mean seems to be around 0.15.

With the longer tail at higher values, the mean of the (skewed) flam distribution is 0.25, while the mode is indeed closer to 0.15. We chose to point out the mean for comparison to studies that did not examine the shape of the distribution (e.g. DeMott 2015), where an average value is reported.

6.) p 8, l 10: Here it is reported that values for flam range from 3 to 73 %. Why are values of 76 and 77 % reported in Fig 2? (difference between ZINC and SPIN?)

Yes; as with previous points we believe the caption change in the figures now clarifies this.

7.) Figure 7: The colors are hard to discern. Four colors are shown in the legend, but several other colors are shown in the graph. This is potentially because overlapping colors result in "new" colors? A different representation would be helpful, e.g. show the four probability distributions not as histograms but as line plots.

We agree with the reviewer and have replaced Figure 7 to show line plots instead of bars.

8.) p 9‐p 10: The ice growth model is insufficiently described. What are the assumptions? How are things calculated? I agree with the other reviewer that this needs considerably more discussion.

We agree and refer Reviewer 2 to the points made by Reviewer 1. We have expanded this section by several paragraphs and have included the relevant references. We believe the model is more clear and the manuscript improved as a result.

9.) p 10, l 13: flam and cf are not equivalent, but rather "inversely equivalent".

"inversely" added

Technical comments

p 7, l 15: "import" "important"

Corrected

p 7, l 21: "middle‐top of the SPIN" → "middle and top section of the SPIN"

Change made

p 11, l 12: "activation curves of droplets. . . "

Corrected

p 11, l 13: "black and blue" "red and blue".

Corrected

p 11, l 18: "4. Conclusions" (section numbering)

Corrected

p12, l 17: "variability to be conducted"

We believe original wording is correct ('to' not needed)

p 27, l 3: "Unlike Figure 6" should read "Unlike Figure 8"

Corrected

p 27, l 15 : "black and blue" "red and blue"

Corrected

Additional References

DeMott, P. J., O. Mohler, O. Stetzer, G. Vali, Z. Levin, M. D. Petters, M. Murakami, T.

Leisner, U. Bundke, H. Klein, Z. A. Kanji, R. Cotton, H. Jones, S. Benz, M. Brinkmann, D. Rzesanke, H. Saathoff, M. Nicolet, A. Saito, B. Nillius, H. Bingemer, J. Abbatt, K. Ardon, E. Ganor, D. G. Georgakopoulos, and C. Saunders (2011), RESURGENCE IN ICE NUCLEI MEASUREMENT RESEARCH, Bulletin of the American Meteorological Society, 92(12), 1623-+, doi:10.1175/bams-d-10-3119.1.

Järvinen, E., Schnaiter, M., Mioche, G., Jourdan, O., Shcherbakov, V. N., Costa, A., Afchine, A., Kraemer, M., Heidelberg, F., Jurkat, T., Voigt, C., Schlager, H., Nichman, L., Gallagher, M., Hirst, E., Schmitt, C., Bansemer, A., Heymsfield, A., Lawson, P., Tricoli, U., Pfeilsticker, K., Vochezer, P., Maehler, O., and Leisner, T.: Quasi-Spherical Ice in Convective Clouds, 73, 3885-3910, doi:10.1175/JAS-D-15-0365.1, 2016. Koop, T., Luo, B., Tsias, A. and Peter, T.: Water activity as the determinant for homogeneous ice nucleation in aqueous solutions, Nature, 406, 611-614, doi:10.1038/35020537, 2000. Magee, N., Moyle, A. M., and Lamb, D.: Experimental determination of the deposition coefficient of small cirrus-like ice crystals near-50 degrees C, Geophys. Res. Lett., 33, doi:10.1029/2006gl026665, 2006. Rogers, R. R., and Yau, M. K.: A Short Course in Cloud Physics, Third ed., International Series in Natural Philosophy, Pergamon Press, Oxford, 1989. Skrotzki, J., Connolly, P., Schnaiter, M., Saathoff, H., Mohler, O., Wagner, R., Niemand, M., Ebert, V., and Leisner, T.: The accommodation coefficient of water molecules on ice - cirrus cloud studies at the AIDA simulation chamber, Atmos. Chem. Phys., 13, 4451-4466, doi:10.5194/acp-13-4451-2013, 2013.